# Analytical and Numerical Treatment of Continuous Ageing in the Voter Model

**DOI:** 10.3390/e24101331

**Published:** 2022-09-21

**Authors:** Joseph W. Baron, Antonio F. Peralta, Tobias Galla, Raúl Toral

**Affiliations:** 1Instituto de Física Interdisciplinar y Sistemas Complejos IFISC (CSIC-UIB), 07122 Palma de Mallorca, Spain; 2Department of Network and Data Science, Central European University, A-1100 Vienna, Austria

**Keywords:** voter model, ageing, non-Markovian systems, thinning algorithm

## Abstract

The conventional voter model is modified so that an agent’s switching rate depends on the ‘age’ of the agent—that is, the time since the agent last switched opinion. In contrast to previous work, age is continuous in the present model. We show how the resulting individual-based system with non-Markovian dynamics and concentration-dependent rates can be handled both computationally and analytically. The thinning algorithm of Lewis and Shedler can be modified in order to provide an efficient simulation method. Analytically, we demonstrate how the asymptotic approach to an absorbing state (consensus) can be deduced. We discuss three special cases of the age-dependent switching rate: one in which the concentration of voters can be approximated by a fractional differential equation, another for which the approach to consensus is exponential in time, and a third case in which the system reaches a frozen state instead of consensus. Finally, we include the effects of a spontaneous change of opinion, i.e., we study a noisy voter model with continuous ageing. We demonstrate that this can give rise to a continuous transition between coexistence and consensus phases. We also show how the stationary probability distribution can be approximated, despite the fact that the system cannot be described by a conventional master equation.

## 1. Introduction

Sociophysics, understood as the application of physics-inspired techniques to problems of sociological interest, has made important progress towards the understanding of consensus formation in populations [1,2,3,4,5,6,7]. In the most basic models, there are two different points of view (or opinions) about a topic. Each individual in the population holds one of these opinions at any one time, and switches between the two states according to basic mechanisms of imitation and spontaneous opinion changes. Imitation plays a similar role to that of positive interaction in physical systems, namely the attractive forces leading particles to align their states, e.g., their spin or velocity. This tends to drive the system towards consensus, or order. The second component in many models of opinion dynamics is a tendency to act independently. This is a force favouring disorder, akin to temperature or noise. The competition between imitation and noise can lead to a phase transition between states near consensus, in which a significant fraction of the population adopts the same opinion, and disordered states in which both opinions coexist in the population with almost equal proportions. This very simple setup lies at the heart of the voter model [8,9,10,11], which we will be considering in detail throughout the paper.

The voter model is an example of an individual-based stochastic model. Other applications of such models include population dynamics in biology [12], game theory and evolution [13], the evolution of languages [14,15,16,17,18], and opinion dynamics more generally [1,19,20]. These models often consist of a very stylised description of the real-world processes, and focus not primarily on studying specific real-world situations, but rather on characterising generic cause-and-effect relations.

One simplifying assumption that is often made in the formulation of such models is that of Markovianity, i.e., the time evolution of the system is taken to depend only on its current state, but not on its previous history. Making this assumption is mathematically convenient, as the machinery for Markovian stochastic processes is well developed [21,22] and can therefore readily be applied.

Discarding memory and history independence is a choice that is made when setting up a particular model. Together with the propensity of physicists to focus on stylised models, the Markovian assumption limits the range of mechanisms and phenomena that can be studied. For example, it is sensible to assume that real-world agents will be subject to some sort of ‘inertia’ when it comes changing their opinion on a particular topic. Their propensity to change will depend on how long they have held their current opinion. Similarly, the rate with which a member of a population infects others with a disease, or recovers from it, will depend on how long they have been infected [23,24,25,26,27]. Similarly, the Markovian assumption breaks down in other applications of individual-based modelling: the production of mRNA and proteins in gene regulatory systems are delayed events [28,29,30,31,32], and anomalous diffusion is manifestly non-Markovian [33,34,35]. Other recent works [36,37] consider memory effects in opinion dynamics happening online, where the whole history of past events of an individual becomes relevant. In this case, the memory effects are a consequence of social online platforms storing personalised information about their users.

There are typically two (related) ways to introduce history dependence into a model. In order to capture the more detailed dynamics leading to delays and memory, one can extend the state space of the model. For example, one can introduce different stages of a disease, or different classes of agents with different ages. The progression between these states can then be modelled as Markovian. Alternatively, one can depart from the assumption of Markovianity entirely and set up a model in which the dynamics become non-Markovian. These two approaches are not entirely disparate; they are merely complementary descriptions of the same underlying processes. Fundamentally, it does not make sense to say that a particular process in nature either has or does not have memory—it is only models of this process that can be Markovian or non-Markovian. Which one it is depends on the scale at which one chooses to model the process mesoscopically.

In previous incarnations of the voter model with ageing [38,39,40,41] (as well as in the majority-vote model [42]), age has been implemented as a set of discrete ‘stages’ through which individuals progress at a constant rate. As they progress through these stages, voters have different proclivities to change opinion. This discretisation of age was a convenient simplification that meant that the usual methods of analysis and simulation for Markovian systems could still be used. Other existing literature, however, addresses models with dependence of the transition rates on internal persistence times [43,44,45]. These latter works relied mostly on numerical simulations.

In this work, on the contrary, we use analytical and numerical methods to study a non-Markovian variant of the voter model, in which each agent has an age (a continuous variable), given by the time since the agent last changed opinions. This setup presents us with several challenges. From a computational standpoint, even a numerical simulation of the system is a non-trivial task, as the celebrated Gillespie algorithm [46] can no longer be used in its standard form. Modifications are instead necessary following the lines of, for example, [47,48,49]. From an analytical perspective, the powerful framework associated with the master equation has to be substantially modified to deal with non-Markovian systems [50,51]. To remedy this, path-integral approaches have been developed that allow one to perform system-size expansions, thus avoiding the need to write a master equation [26,28,33]. Alternatively, there often exist parameter regimes in which one can approximate the full non-Markovian dynamics with an effective Markovian system [23,24,39,52]. Such an approximation replaces history dependence with an additional non-linearity.

We demonstrate several advantages to employing a continuous ageing process over a discrete version. In addition to being more natural and realistic, the elimination of the age variables for each individual to obtain the macroscopic concentrations for either opinion as a whole is far simpler in the continuous case. Further, we are able to demonstrate that the approach to consensus in the case of power law ageing and in the absence of spontaneous opinion changes follows a fractional differential equation. This is not at all transparent for discrete ageing. Moreover, we are able to approximate the full stationary distribution using an adiabatic reduction method. All of these analytical advantages are available without sacrificing computational efficiency through the appropriation of the thinning algorithm by Lewis and Shedler [49,53].

The rest of the work is structured as follows. We first introduce the modified voter model with continuous ageing in Section 2.1. We then describe how the thinning algorithm can be used to simulate the system in Section 2.2. Section 3.1 focuses on the model without spontaneous opinion changes. In particular, we use a deterministic approximation to examine the approach to consensus and the conditions under which consensus is reached in Section 3.2. This is achieved through a linearisation of the deterministic dynamics near an absorbing state. We analyse different scenarios of ageing and show that the type of ageing can severely affect the presence or absence of ordering, and, in cases where the system orders, the approach to consensus. In Section 3.3.1, we include the possibility of a spontaneous change of opinion and analyse the resulting continuous phase transition between order and disorder, and we describe how ageing makes this transition different from that in the conventional voter model. Finally, we go beyond the deterministic dynamics and analyse the stochastic fluctuations about the deterministic fixed points in Section 3.3.2. We discuss the results and conclude in Section 4.

## 2. Methods

### 2.1. Voter Model with Age-Dependent Switching Rates

Consider a population of *N* agents (voters). At each point in time *t*, each agent is described by a binary variable $s_{i}\left(t\right)=\pm 1$, representing the agent’s opinion, where i=1,⋯,N. We write $n^{\pm }\left(t\right)$ for the total number of voters in each of the two states at time *t*. We have $n^{+}\left(t\right)+n^{-}\left(t\right)=N$ for all *t*. The corresponding intensive variables are $x^{\pm }\left(t\right)=n^{\pm }\left(t\right)/N$, and we have $x^{+}\left(t\right)+x^{-}\left(t\right)=1$. Adopting the terminology of ferromagnetism, it is convenient to introduce a ‘magnetisation’ $m\left(t\right)=|x^{+}\left(t\right)-x^{-}\left(t\right)|$.

The variables $s_{i}\left(t\right)$ evolve in time via a stochastic process, defined by the rate $r_{i}\left(t\right)$ at which agent *i* changes its current opinion from $s_{i}$ to $-s_{i}$. In addition to the state of the system $s\left(t\right)=\{s_{1}\left(t\right),\cdots ,s_{N}\left(t\right)\}$, the rate $r_{i}\left(t\right)$ can depend on the age $\tau_{i}\left(t\right)$ of the agent at time *t*; in our model, this is defined as the time since the agents’ last change of state. Other names given to this ‘age’ variable appear in the literature as ‘persistence time’ [43] or ‘internal time’ [40,44]. Given the all-to-all connectivity of agents that we adopt throughout this paper, the rates are
(1)$$\begin{array}{c} r_{i}\left(t\right)\equiv r\left[s_{i}\left(t\right)\to -s_{i}\left(t\right);s\left(t\right);\tau_{i}\left(t\right)\right]=a+\frac{p_{\tau_{i}\left(t\right)}}{N}\sum_{j\neq i}\left[1-\delta_{s_{i}\left(t\right),s_{j}\left(t\right)}\right]. \end{array}$$ The first term, involving the model parameter *a*, describes spontaneous changes of opinion. The second term reflects the imitation mechanism. This term is proportional to the density of agents holding the opinion opposite to the current state of agent *i*. The factor $p_{\tau_{i}\left(t\right)}$, the activation rate, represents the age dependence of the propensity of agent *i* to imitate another agent (we will refer to this as the ‘ageing profile’). The rates in Equation (Equation 1) can also be re-expressed in the following form:(2)$$\begin{array}{c} r_{\tau }^{\pm }\left(t\right)=a+p_{\tau }x^{\pm }\left(t\right). \end{array}$$ These are the per capita rates with which voters of age τ switch from state ∓1 to ±1.

Classifying agents by their age τ, we define the number densities $n^{\pm }\left(\tau ,t\right)$ such that $\int_{\tau_{1}}^{\tau_{2}}d\tau \;n^{\pm }\left(\tau ,t\right)$ is the number of voters in opinion state ±1 with ages between $\tau_{1}$ and $\tau_{2}$ at time *t*. The corresponding intensive variables are $x^{\pm }\left(\tau ,t\right)=n^{\pm }\left(\tau ,t\right)/N$, and we have the relation
(3)$$x^{\pm }\left(t\right)=\int_{0}^{\infty }d\tau \;x^{\pm }\left(\tau ,t\right),$$
and similarly for $n^{\pm }\left(t\right)$.

We first note that the model reduces to the conventional noisy voter model [54,55] when $p_{\tau }$ is a constant not depending on τ. The phenomenology of this special case is well known: for a=0 (and assuming *N* is finite), the system reaches a consensus state in which all agents hold the same opinion (either +1 or −1). One then has m(t)→1 in every realisation as t→∞. For a>0, there is a critical value $a=a_{c}$ such that m(t) assumes values close to 0 for $a>a_{c}$. Here, as t→∞, both opinions are present in the population in similar proportions at any one time. However, when $a<a_{c}$, m(t) tends to a non-zero value. The critical value $a_{c}=O\left(1/N\right)$ separates the two regimes. We note that $a_{c}\to 0$ as N→∞; hence, this is a finite-size transition. As we will show, both the approach to consensus (for a=0) and the mean magnetisation of the system (for a>0) can be affected dramatically by the introduction of different ageing profiles $p_{\tau }$.

If the activation rate is a decreasing function such that the asymptotic value is lower than the initial one, $p_{\tau =\infty }<p_{\tau =0}$, we have a situation in which it becomes more difficult for an agent to change state the longer it has been holding the current state. This is typical of ageing, in which an agent becomes more reluctant to adopt new opinions. This increased resistance to changing state has also been described as ‘increasing inertia’ [43]. The opposite case, $p_{\infty }>p_{0}$, which we name anti-ageing, models a situation in which agents become tired of the current state and are more prone to adopting new viewpoints. In previous studies of the discrete version of ageing [38,39], it was shown that when $p_{\infty }=0$, the system may or may not reach consensus. This depends on whether the decay of $p_{\tau }$ to zero is slower or faster, respectively, than 1/τ. If, on the other hand, the activation rate tends to a non-zero value from above $p_{0}>p_{\infty }>0$, the system orders with an exponential decay of the fraction $x^{\pm }\left(t\right)$ of individuals to the consensus value. Finally, if the activation rate tends to a non-zero value from below (anti-ageing) $p_{\infty }>p_{0}>0$, then the system does not reach consensus. As explained in Ref. [43], the results for $p_{\infty }>0$ can be understood heuristically by an amplification by the ageing mechanism of any small asymmetry in the initial conditions.

### 2.2. Modified Thinning Algorithm for Simulation

The fact that the rates $r_{i}\left(t\right)$ in Equation (Equation 1) depend on age (and hence vary with time, even when the configuration s(t) remains constant) means that we cannot use the traditional Gillespie algorithm [46] to carry out individual-based simulations. One strategy to overcome this issue would be to employ a staged transition from one state to the other with Markovian progression between the stages [38,39,40]. This is the approach of defining an alternative Markovian dynamics in a space with additional degrees of freedom, as discussed in the Introduction.

A perhaps more elegant solution is to use a modification of the so-called ‘thinning’ algorithm by Lewis and Shedler [49,53], which we will now describe.

The total rate with which any switching event occurs in the population, $R\left(t\right)=\sum_{i}r_{i}\left(t\right)$, is time-dependent. One notes that the probability that voter *i* changes its opinion, *given* that a reaction occurs at time *t*, is $P_{i}\left(t\right)=r_{i}\left(t\right)/R\left(t\right)$. Lewis and Shedler’s insight was to add an additional ‘null event’, with a time-dependent rate chosen such that the total rate of events (actual events and the null event) is constant in time. This is possible provided that R(t) is bounded from above (we discuss this for our system below). Imagine that we have reached a time $t_{0}$ in our simulations. We then introduce an auxiliary reaction occurring with rate $R_{0}\left(t\right)=R_{\max}\left(t_{0}\right)-R\left(t\right)$, where $R_{\max}\left(t_{0}\right)\geq \max_{t>t_{0}}R\left(t\right)$. When this additional reaction occurs, nothing happens in the population. The total reaction rate until the next event is now constant in time and equal to $R_{\max}\left(t_{0}\right)$. The individual rates $\left(R_{0}\left(t\right),\left\{r_{i}\left(t\right)\right\}\right)$ are time-dependent, and preserve the statistical properties of the original process.

One requirement for this construction is that the total rate R(t) must be bounded from above, $R\left(t\right)\leq R_{\max}\left(t_{0}\right)$ for all $t>t_{0}$. If the condition $p_{\tau }<p_{\max}$ for all τ is satisfied for some $p_{\max}$, then we can bound $R\left(t\right)\leq R_{\max}\left(t_{0}\right)\equiv aN+2p_{\max}n^{-}\left(t_{0}\right)n^{+}\left(t_{0}\right)/N$. The condition $p_{\tau }<p_{\max}$ is naturally satisfied for a decreasing function $p_{\tau }$ of age τ. In the event that $p_{\tau }$ were unbounded, one could use the alternative approximate simulation method in Ref. [48].

More precisely, we simulate the system as follows:Set t=0. Initialise all ages $\tau_{i}\left(0\right)=0$ and draw the states $s_{i}\left(0\right)$ from the desired initial distribution.Assume that the simulation has reached time *t*. Set $R_{\max}=aN+2p_{\max}n^{-}\left(t\right)n^{+}\left(t\right)/N$.Draw a uniform random number u from the interval (0,1] and calculate the time interval to the next event $\Delta =-\ln\;u/R_{\max}$. Update time and the ages of all voters such that t→t+Δ, and $\tau_{i}\to \tau_{i}+\Delta$ for i=1,⋯,N.Choose the type of event to occur (all rates are evaluated at the updated time):(i)With probability $R_{0}\left(t\right)/R_{\max}$, nothing happens.(ii)With probability $r_{i}\left(t\right)/R_{\max}$, voter *i* switches opinion, $s_{i}\left(t\right)\to -s_{i}\left(t\right)$; set $\tau_{i}\left(t\right)=0$; update $n^{\pm }\left(t\right)$.Go to item 2.

One inevitable consequence of the introduction of ageing into the model is that individuals become distinguishable; each individual transitions with a different rate. This means that there are *N* possible events to consider, rather than simply two events, as would be the case for the voter model without ageing. The above algorithm comes as close to replicating the efficiency of the Gillespie algorithm as is feasible, given the far greater number of possible events to consider.

## 3. Results

### 3.1. Deterministic Approximation for the Model without Spontaneous Opinion Changes (a=0)

We now set out to determine analytically how continuous ageing affects the approach to consensus in the model without spontaneous opinion changes (a=0). We examine the model with spontaneous changes of opinion (a>0) in Section 3.3.

We could, in principle, begin by writing the generating functional [26,33] using the rates in Equation (Equation 2) and then perform a system-size expansion to obtain a systematic approximation to the individual-based dynamics. With that being said, we suppose for now that *N* is large enough that fluctuations (of order $1/\sqrt{N}$) in the numbers of voters in either state can be ignored and we simply aim at writing deterministic rate equations for the concentrations $x^{\pm }\left(t\right)=n^{\pm }\left(t\right)/N$. This approach is valid in the thermodynamic limit N→∞. We relax the deterministic assumption in Section 3.3, where we examine stochastic fluctuations about deterministic fixed points.

Let us consider a small interval of time Δt. At time *t*, the average number of individuals in the state ±1 and ages in the interval [τ,τ+Δt) is $\Delta tNx^{\pm }\left(\tau ,t\right)$. Following Equation (Equation 2), during the time interval [t,t+Δt), each one of these agents switches state with probability $\Delta t\;p_{\tau }\;x^{\mp }\left(t\right)$. Noting that the number of voters in state ±1 with ages in [τ+Δt,τ+2Δt) at time t+Δt is equal to the number of voters in the same state at time *t* with ages in [τ,τ+Δt) minus the number of those voters that change state in the interval [t,t+Δt), we find
(4)$$\begin{array}{c} N\Delta tx^{\pm }\left(\tau +\Delta t,t+\Delta t\right)=N\Delta tx^{\pm }\left(\tau ,t\right)-\left[\Delta tNx^{\pm }\left(\tau ,t\right)\right]\times \left[\Delta t\;p_{\tau }\;x^{\mp }\left(t\right)\right]. \end{array}$$ Expanding the left-hand side to the first order in Δt and taking limit Δt→0, we obtain
(5)$$\begin{array}{c} \frac{\partial x^{\pm }\left(\tau ,t\right)}{\partial t}+\frac{\partial x^{\pm }\left(\tau ,t\right)}{\partial \tau }=-p_{\tau }x^{\mp }\left(t\right)x^{\pm }\left(\tau ,t\right). \end{array}$$ This equation has to be implemented with an initial condition $x^{\pm }\left(\tau ,t=0\right)$. If all ages are set to τ=0 at t=0, then the initial condition is
(6)$$x^{\pm }\left(\tau ,t=0\right)=x_{0}^{\pm }\delta \left(\tau \right),$$
where $x_{0}^{\pm }=x^{\pm }\left(t=0\right)$ are the initial proportions of voters holding the ±1 opinion, and δ(·) is the Dirac-delta function. Further, the average number of voters arriving at the state ±1 during the time interval [t,t+Δt) and subsequently adopting age τ=0 can be approximated by $\Delta tNx^{\pm }\left(t\right)\int_{0}^{t}du\;p_{u}\;x^{\mp }\left(u,t\right)$. The number density of agents with zero age is hence $n^{\pm }\left(\tau =0,t\right)=Nx^{\pm }\left(t\right)\int_{0}^{t}du\;p_{u}\;x^{\mp }\left(u,t\right)$. This encapsulates the influx of age-zero voters into state ±1 due to opinion changes at time *t*, and translates into the boundary condition
(7)$$\begin{array}{c} x^{\pm }\left(\tau =0,t\right)=x^{\pm }\left(t\right)\int_{0}^{t}du\;p_{u}x^{\mp }\left(u,t\right). \end{array}$$ Equation (Equation 5) along with the initial condition in Equation (Equation 6) and the boundary condition in Equation (Equation 7) are the basis of our analysis.

Following [33,56,57,58], we use the method of characteristics to solve Equation (Equation 5). Parameterising the coordinates *t* and τ such that
(8)$$\begin{array}{c} \tau =s,\;t=c+\tau , \end{array}$$
one finds through the characteristics of constant *c* that
(9)$$\begin{array}{cc} \frac{dx^{\pm }\left(s,s+c\right)}{ds} & =\frac{\partial x^{\pm }\left(\tau ,t\right)}{\partial t}\frac{dt}{ds}+\frac{\partial x^{\pm }\left(\tau ,t\right)}{\partial \tau }\frac{d\tau }{ds}=\frac{\partial x^{\pm }\left(\tau \left(s\right),t\left(s\right)\right)}{\partial t(s)}+\frac{\partial x^{\pm }\left(\tau \left(s\right),t\left(s\right)\right)}{\partial \tau (s)}\\ \; & =-p_{s}x^{\mp }\left(s+c\right)x^{\pm }\left(s,s+c\right). \end{array}$$ This can be solved to yield
(10)$$\begin{array}{c} x^{\pm }\left(s,s+c\right)=x^{\pm }\left(0,c\right)e^{-\int_{0}^{s}du\;p_{u}x^{\mp }\left(c+u\right)}. \end{array}$$ We then obtain, in the original coordinates *t* and τ,
(11)$$\begin{array}{cc} x^{\pm }\left(\tau ,t\right) & =x^{\pm }\left(0,t-\tau \right)e^{-\int_{0}^{\tau }ds\;p_{s}x^{\mp }\left(t-\tau +s\right)},\;t\geq \tau . \end{array}$$ Note that this expression is not yet a closed solution since $x^{\mp }\left(\cdot \right)$ appears on the right-hand side. This quantity must satisfy Equation (Equation 3). The factor $x^{\pm }\left(0,t-\tau \right)$ in front of the exponential is given by the boundary condition in Equation (Equation 7).

The expression in Equation (Equation 11) has a simple interpretation. The right-hand side accounts for voters that switch at an earlier time t−τ (and hence become age-zero agents at that time), and who then survive τ units of time without switching opinions again. The exponential factor in Equation (Equation 11) is the probability that such a voter does not change opinion in this time interval.

We can also integrate Equation (Equation 5) directly with respect to τ and use Equations (Equation 3) and (Equation 7) to obtain an integral-differential equation for the evolution of $x^{\pm }\left(t\right)$:(12)$$\begin{array}{c} \frac{dx^{\pm }\left(t\right)}{dt}=x^{\pm }\left(t\right)\int_{0}^{\infty }d\tau \;p_{\tau }x^{\mp }\left(\tau ,t\right)-x^{\mp }\left(t\right)\int_{0}^{\infty }d\tau \;p_{\tau }x^{\pm }\left(\tau ,t\right). \end{array}$$ The initial condition in Equation (Equation 6) implies $x^{\pm }\left(\tau >t,t\right)=0$ (no agent can have an age larger than the current time *t*), so that the upper limits of the integrals can be replaced by *t*,
(13)$$\begin{array}{c} \frac{dx^{\pm }\left(t\right)}{dt}=x^{\pm }\left(t\right)\int_{0}^{t}d\tau \;p_{\tau }x^{\mp }\left(\tau ,t\right)-x^{\mp }\left(t\right)\int_{0}^{t}d\tau \;p_{\tau }x^{\pm }\left(\tau ,t\right). \end{array}$$ This is to be implemented with the initial condition Equation (Equation 6), which implies $x^{\pm }\left(t=0\right)=x_{0}^{\pm }$. The first term on the right-hand side of Equation (Equation 13) describes the influx of voters newly converted to the ±1 opinion. The second term represents the outflux of voters, namely those that are converted to the ∓1 opinion. Again, these equations are not closed in terms of $x^{\pm }\left(t\right)$. Nevertheless, as discussed in the next section, they can be used as a starting point to analyse the approach to the consensus states $x^{+}=0$ or $x^{+}=1$.

### 3.2. Ordering Dynamics with Continuous Ageing

In the conventional voter model without ageing, consensus is reached through fluctuations, i.e., there is no deterministic drift driving the system to absorption. As we show below, the system with a power-law ageing profile $p_{\tau }$ instead moves deterministically towards consensus. On the other hand, when the ageing profile decays exponentially, the system tends towards a frozen state which is dependent on the initial configuration of opinions. In the case where a constant value of the switching propensity $p_{\tau }\to p_{\infty }$ is approached as τ→∞, we show instead that the ultimate fate of the system is determined by whether $p_{\tau }$ is an increasing or decreasing function of τ. If $p_{0}>p_{\infty }$, then the system approaches consensus. If $p_{0}<p_{\infty }$, consensus is not achieved.

These observations were also made in the case of discrete ageing [38]. However, our objective here is to demonstrate the relative ease with which the results can be derived in the model with continuous ageing. For the power-law ageing profile, we demonstrate additionally that the ensemble-averaged number of agents in a specific opinion satisfies a fractional differential equation. We also verify our results numerically with the aforementioned thinning algorithm.

#### 3.2.1. Linearisation Close to the Absorbing State

We now examine the behaviour close to the absorbing state of consensus at $x^{-}=1$ (and $x^{+}=0$). A similar analysis could be performed for the state $x^{-}=0$ (and $x^{+}=1$). Here, we assume that $x^{+}\left(t\right)$ is a small quantity and examine its time dependence. With this in mind, we can linearise Equations (Equation 11) and (Equation 12) in terms of $x^{+}\left(t\right)$. Our aim is then to eliminate the age variable τ from these linearised equations in order to find a closed equation for $x^{+}\left(t\right)$.

To carry out the analysis, we start at time t=0 with $x_{0}^{+}=\epsilon$ and $x^{+}\left(\tau ,0\right)=\epsilon \delta \left(\tau \right)$. We then have $x^{-}\left(0\right)=1-\epsilon$, and we set $x^{-}\left(\tau ,0\right)=\left(1-\epsilon \right)\delta \left(\tau \right)$. We assume ϵ≪1 and—within a linear expansion—that $x^{+}\left(t\right)$ and $x^{+}\left(\tau ,t\right)$ remain of order ϵ, and we neglect terms of order $\epsilon^{2}$ and higher.

We focus on the expression for $x^{+}\left(\tau ,t\right)$ resulting from Equation (Equation 11). Within the linear expansion in ϵ, we can replace $x^{-}\left(t-\tau +s\right)=1$ in the exponential. Reiterating that $x^{+}\left(\tau ,t\right)=O\left(\epsilon \right)$, we then obtain
(14)$$\begin{array}{c} x^{+}\left(\tau ,t\right)=x^{+}\left(0,t-\tau \right)\Psi \left(\tau \right)+O\left(\epsilon^{2}\right), \end{array}$$
with the survival probability $\Psi \left(t\right)=e^{-\int_{0}^{t}d\tau \;p_{\tau }}$.

We next write $p_{\tau }=\psi \left(\tau \right)/\Psi \left(\tau \right)$, where $\psi \left(t\right)=-\frac{d\Psi (t)}{dt}$ is the probability density function of switching times for an agent who is of age zero at time t=0. As shown in Appendix A, the second term of the right-hand side of Equation (Equation 13) can then be written as
(15)$$\begin{array}{cc} \int_{0}^{t}d\tau \;p_{\tau }x^{+}\left(\tau ,t\right) & =\int_{0}^{t}d\tau \;K\left(\tau \right)x^{+}\left(t-\tau \right), \end{array}$$
where the memory kernel K(τ) is defined via its Laplace transform $\hat{K}\left(u\right)=\hat{\psi }\left(u\right)/\hat{\Psi }\left(u\right)$.

For the first integral in the right-hand side of Equation (Equation 13), we use the fact that the number of agents in state +1 is of order ϵ, within our approximation. The fraction of agents that can switch out of state −1 is then also (at most) of order ϵ. Noting that all agents have age zero at the beginning, the age of agents who remain in the state −1 up to time *t* is *t*, and we have $x^{-}\left(\tau ,t\right)=\left(1-\epsilon \right)\delta \left(t-\tau \right)+O\left(\epsilon \right)$. The first term describes the agents that do not change state, and there is an O(ϵ) correction from agents who do change state. Hence,
(16)$$\begin{array}{c} \int_{0}^{t}d\tau \;p_{\tau }x^{-}\left(\tau ,t\right)=p_{t}+O\left(\epsilon \right). \end{array}$$ Replacing $x^{-}\left(t\right)=1+O\left(\epsilon \right)$ and the above results in Equation (Equation 13), we finally arrive at
(17)$$\begin{array}{c} \frac{dx^{+}}{dt}=p_{t}x^{+}\left(t\right)-\int_{0}^{t}d\tau \;K\left(\tau \right)x^{+}\left(t-\tau \right). \end{array}$$ We have thus achieved our goal of eliminating the age coordinates in favour of the global variable $x^{+}\left(t\right)$. One notes that in place of this single equation, the linearisation in Ref. [38] gave rise to a pair of coupled equations, one governing the influx of voters into a particular opinion and the other governing the total number in that opinion.

#### 3.2.2. Power-Law Ageing and the Approach to Consensus

Let us now consider a particular example of the approach to consensus in the presence of ageing. Consider the power-law ageing profile
(18)$$\begin{array}{c} p_{\tau }=\frac{\gamma }{t_{0}+\tau }, \end{array}$$
with constants $\gamma >0,t_{0}>0$. The corresponding survival function and distribution of switching times are, respectively,
(19)$$\begin{array}{c} \Psi \left(\tau \right)={\left(1+\frac{\tau }{t_{0}}\right)}^{-\gamma },\;\;\;\psi \left(\tau \right)=\frac{\gamma }{t_{0}}\;{\left(1+\frac{\tau }{t_{0}}\right)}^{-(1+\gamma )}. \end{array}$$ Here, we consider only the case 0<γ<1 (see Appendix B for a discussion of the case of γ∈N). **-** **Fractional differential equation**

We now show that the concentration of voters in state +1 can be approximated by a fractional differential equation upon the approach to consensus. We first compute the Laplace transform of Ψ(τ) as
(20)$$\hat{\Psi }\left(u\right)=t_{0}e^{t_{0}u}{\left(t_{0}u\right)}^{-1+\gamma }\Gamma \left[1-\gamma ,t_{0}u\right],$$
where Γ(·,·) is the incomplete Gamma function. For γ<1, we now investigate the limit of small *u*. One finds, to leading order,
(21)$$\begin{array}{c} \hat{K}\left(u\right)\approx \frac{u^{1-\gamma }}{T^{\gamma }}, \end{array}$$
where $T^{\gamma }=\Gamma \left(1-\gamma \right)t_{0}^{\gamma }$, and Γ(·) is the standard gamma function.

It is now useful to recall that the Riemann–Liouville fractional derivative of a function f(t) is defined as [35]
(22)$$\begin{array}{c} {}_{0}D_{t}^{1-\gamma }f\left(t\right)=\frac{1}{\Gamma (\gamma )}\frac{\partial }{\partial t}\int_{0}^{t}dt^{\prime }\;\frac{f(t^{\prime })}{{\left(t-t^{\prime }\right)}^{1-\gamma }}, \end{array}$$
and that it has the following Laplace transform
(23)$$\begin{array}{c} L\left[{}_{0}D_{t}^{1-\gamma }f\left(t\right)\right]=u^{1-\gamma }\hat{f}\left(u\right). \end{array}$$ Taking the Laplace transform of the second term on the right-hand side of Equation (Equation 17), substituting the expression for the memory kernel in Equation (Equation 21), and using (Equation 23), one then finds, after transforming back to *t*,
(24)$$\begin{array}{c} \frac{dx^{+}}{dt}\approx \frac{\gamma x^{+}\left(t\right)}{t}-\frac{1}{t_{0}^{\gamma }\Gamma \left(1-\gamma \right)}{}_{0}D_{t}^{1-\gamma }\left[x^{+}\left(t\right)\right]. \end{array}$$ This fractional differential equation is valid for $t\gg t_{0}$.**-** **Asymptotic behaviour**

We now deduce the scaling of $x^{+}\left(t\right)$ for large times. We multiply both sides of Equation (Equation 24) (or equivalently of Equation (Equation 17)) by *t*, and then take a Laplace transform. Using the fact that $L\left[tf(t)\right]=-\frac{d}{du}\hat{f}\left(u\right)$, one obtains, after some algebra,
(25)$$\begin{array}{c} \left[1+\gamma +\frac{\left(1-\gamma \right)}{t_{0}^{\gamma }\Gamma \left(1-\gamma \right)}u^{-\gamma }\right]{\hat{x}}^{+}+\left[u+\frac{u^{1-\gamma }}{t_{0}^{\gamma }\Gamma \left(1-\gamma \right)}\right]\frac{d{\hat{x}}^{+}}{du}=0. \end{array}$$ This can be solved to yield
(26)$$\begin{array}{c} {\hat{x}}^{+}\left(u\right)=C\frac{u^{\gamma -1}}{1+t_{0}^{\gamma }\Gamma \left(1-\gamma \right)u^{\gamma }}, \end{array}$$
where *C* is a constant. For small *u*, we therefore have ${\hat{x}}^{+}\left(u\right)∼u^{\gamma -1}$. Using the Tauberian theorem [58,59,60],
(27)$$\begin{array}{c} \hat{x}\left(u\right)∼u^{\gamma -1}\Leftrightarrow x\left(t\right)∼t^{-\gamma }, \end{array}$$
we therefore find that $x^{+}\left(t\right)∼t^{-\gamma }$ at long times. This scaling behaviour is verified in Figure 1.

#### 3.2.3. Ageing Profile with Non-Zero Asymptotic Value

We now consider an ageing profile that tends to a finite value at infinite age,
(28)$$\begin{array}{c} p_{\tau }=p_{\infty }+\frac{\gamma }{t_{0}+\tau }, \end{array}$$
with constants $p_{\infty }>0,t_{0}>0$. From this definition, we note that $\gamma =\left(p_{0}-p_{\infty }\right)t_{0}$.

If γ>0, then $p_{\tau }$ is a decreasing function of τ as $p_{\infty }<p_{0}$. In this case, as in the ones considered before, agents become more ossified in their views if they maintain their opinions. However, in contrast to the previous cases, there now remains a residual propensity for change even at very large ages τ. If, on the other hand, $p_{\infty }>p_{0}$, i.e., γ<0 (but also $\gamma >-p_{\infty }t_{0}$ to ensure that $p_{\tau }>0,\;\forall \tau$), then the propensity to change state increases with age (but is limited from above by the finite value $p_{\infty }$). We call this ‘anti-ageing’.

To analyse if consensus is reached for this functional form of the ageing profile, we take Equation (Equation 17) as a starting point. The kernel $K\left(\tau \right)=L^{-1}\left[1/\hat{\Psi }\left(u\right)-u\right]$ can be computed from the knowledge of the Laplace transform of $\Psi \left(t\right)=e^{-p_{\infty }t}{\left(1+t/t_{0}\right)}^{-\gamma }$
(29)$$\hat{\Psi }\left(u\right)=t_{0}e^{t_{0}\left(u+p_{\infty }\right)}{\left[t_{0}\left(u+p_{\infty }\right)\right]}^{-1+\gamma }\Gamma \left[1-\gamma ,t_{0}\left(u+p_{\infty }\right)\right].$$ In order to find the asymptotic solution of Equation (Equation 17), we replace $p_{t}$ in the first term on the right-hand side of the equation by its asymptotic value $p_{\infty }$. We note that we retain the dependence of the ageing profile on γ in the second term (through the expression in Equation (Equation 29)). Upon taking the Laplace transform of this equation and using $L\left[\frac{dx^{+}\left(t\right)}{dt}\right]=u{\hat{x}}^{+}\left(u\right)-x^{+}\left(0\right)$, we find, after some algebra, that
(30)$${\hat{x}}^{+}\left(u\right)=\frac{x(0)}{\frac{1}{\hat{\Psi }\left(u\right)}-p_{\infty }}.$$ A detailed analysis of this equation shows that the right-hand side has a single pole at a value $u^{*}$ whose position depends on the value of γ. Inverting the Laplace transform, the pole translates into an asymptotic behaviour $x^{+}\left(t\right)∼e^{u^{*}t}$. Furthermore, it can be shown, as illustrated in Figure 2 for a given choice of the remaining model parameters, that the value of $u^{*}$ is negative for γ>0 (ageing) and positive for γ<0 (anti-ageing). Therefore, we conclude that the ageing mechanism with a final non-zero value of the activation probability leads to consensus only when this final value is approached from above. On the other hand, a situation of disorder (lack of consensus) is obtained when the final value of $p_{\tau }$ is approached from below. This counter-intuitive result agrees with the results observed in the discrete version of the model [38]. We show in Figure 3 the decay of the density $x^{+}\left(t\right)$ in the particular case $t_{0}=0.8,\;p_{\infty }=0.5,\;\gamma =0.1$, together with the exponential functional form $e^{u^{*}t}$.

#### 3.2.4. Exponential Ageing and the Frozen State

To demonstrate how severely the qualitative behaviour of the deterministic dynamics can be affected by the precise choice of ageing profile, we now consider the case of exponential ageing,
(31)$$\begin{array}{c} p_{\tau }=p_{0}e^{-\tau /t_{0}}. \end{array}$$ The corresponding survival function is $\Psi \left(\tau \right)=\exp\left[-p_{0}t_{0}\left(1-e^{-\tau /t_{0}}\right)\right]$, and the distribution of survival times is $\psi \left(\tau \right)=p_{0}e^{-\tau /t_{0}}\times \exp\left[-p_{0}t_{0}\left(1-e^{-\tau /t_{0}}\right)\right]$.

In this special case, Equation (Equation 17) becomes, after taking the Laplace transform and multiplying through $\hat{\Psi }\left(u\right)$ (and noting that $1-u\hat{\Psi }\left(u\right)=\psi \left(u\right)$ and that $L\left\{e^{-t/t_{0}}x^{+}\left(t\right)\right\}={\hat{x}}^{+}\left(u+1/t_{0}\right)$),
(32)$$\begin{array}{c} {\hat{x}}^{+}\left(u\right)=x^{+}\left(0\right)\hat{\Psi }\left(u\right)+p_{0}\hat{x}\left(u+1/t_{0}\right)\hat{\Psi }\left(u\right). \end{array}$$ One can expand this expression as an infinite series to obtain
(33)$$\begin{array}{c} \hat{x}\left(u\right)=x^{+}\left(0\right)\left[\hat{\Psi }\left(u\right)+p_{0}\hat{\Psi }\left(u\right)\hat{\Psi }\left(u+1/t_{0}\right)+p_{0}^{2}\hat{\Psi }\left(u\right)\hat{\Psi }\left(u+1/t_{0}\right)\hat{\Psi }\left(u+2/t_{0}\right)+\cdots \right]. \end{array}$$ One can also find a series solution for $\hat{\Psi }\left(u\right)$ by first expanding Ψ(t) as a series in $e^{-t/t_{0}}$. One finds
(34)$$\begin{array}{c} \hat{\Psi }\left(u\right)=e^{-p_{0}t_{0}}\left[\frac{1}{u}+\frac{p_{0}t_{0}}{1!}\frac{t_{0}}{1+ut_{0}}+\frac{{\left(p_{0}t_{0}\right)}^{2}}{2!}\frac{t_{0}}{2+ut_{0}}+\frac{{\left(at_{0}\right)}^{3}}{3!}\frac{t_{0}}{3+ut_{0}}\cdots \right]. \end{array}$$

If we now define
(35)$$\begin{array}{c} f_{n}\left(p_{0}t_{0}\right)=\frac{1}{n}+\frac{p_{0}t_{0}}{1!}\frac{1}{1+n}+\frac{{\left(p_{0}t_{0}\right)}^{2}}{2!}\frac{1}{2+n}+\frac{{\left(p_{0}t_{0}\right)}^{3}}{3!}\frac{1}{3+n}\cdots , \end{array}$$
we obtain an expression for the final value $x^{+}\left(\infty \right)$ as a series entirely in the dimensionless parameter $p_{0}t_{0}$ by using the final value theorem for Laplace transforms [61]
(36)$$\begin{array}{c} x^{+}\left(\infty \right)=\lim_{u\to 0}u\hat{x}\left(u\right)=x^{+}\left(0\right)e^{-p_{0}t_{0}}\left[1+p_{0}t_{0}e^{-p_{0}t_{0}}f_{1}+{\left(p_{0}t_{0}\right)}^{2}e^{-2p_{0}t_{0}}f_{1}f_{2}+\cdots \right], \end{array}$$
where $f_{n}$ is shorthand for $f_{n}\left(p_{0}t_{0}\right)$.

We therefore see that when the rate of switching decays exponentially, consensus never occurs on the deterministic level. One can thus find $x^{+}\left(\infty \right)$ as a function of the rate parameters and the starting value. We verify the expression for the frozen state in Equation (Equation 36) in Figure 4. Based on these results, we expect that the faster the decay rate, the larger the value of $x^{+}\left(\infty \right)/x^{+}\left(0\right)$.

### 3.3. Model with Spontaneous Opinion Changes (a>0)

Having studied the effect that continuous ageing has on the approach to consensus, we now allow for the possibility of spontaneous changes of opinion. Without ageing, this is known in the literature as the ‘noisy voter’ or Kirman model [54,55,62]. In other words, now, each voter changes its opinion with a constant rate a≠0 (see Equation (Equation 2)), as well as copying others at an age-dependent rate $p_{\tau }n^{\mp }/N$.

We demonstrate here how the order–disorder transition in the noisy voter model is modified in two different ways. First, the transition in the noisy voter model is discontinuous, in the sense that the mode of the stationary distribution has a discontinuity at the transition. As we will show, there is no such discontinuity in the models with ageing that we consider. Second, the transition occurs at a value of *a* that scales as $N^{-1}$ in the noisy voter model without ageing—that is, this is a finite-size transition. In the model with ageing, instead, the value of *a* at which the transition occurs does not depend on the system size *N*, and remains non-zero as N→∞. Similar results were also derived in Refs. [39,40] for the case of staged ageing. We show here that modelling ageing as a continuous process makes the analysis more straightforward than in the case of staged ageing.

In addition to studying the transition on a deterministic level, we also show how the fluctuations about the deterministic fixed points can be quantified. We show that the stationary distribution can be approximated using an adiabatic elimination [39].

#### 3.3.1. Continuous Phase Transition

Let us now revise the deterministic rate equation in Equations (Equation 5) and (Equation 7) to include spontaneous changes of opinion. One obtains
(37)$$\begin{array}{cc} \frac{\partial x^{\pm }\left(\tau ,t\right)}{\partial \tau }+\frac{\partial x^{\pm }\left(\tau ,t\right)}{\partial t} & =-\left[a+p_{\tau }x^{\mp }\left(t\right)\right]x^{\pm }\left(\tau ,t\right) \end{array}$$
with initial and boundary conditions
(38)$$\begin{array}{cc} x^{\pm }\left(\tau ,t=0\right) & =x^{\pm }\left(0\right)\delta \left(\tau \right),\\ x^{\pm }\left(\tau =0,t\right) & =\int_{0}^{t}d\tau \;\left[a+p_{\tau }x^{\pm }\left(t\right)\right]x^{\mp }\left(\tau ,t\right). \end{array}$$ Now, we suppose that there exists a stationary solution such that $\frac{\partial x{\left(\tau ,t\right)}^{\pm }}{\partial t}=0$, and we write $x^{\pm }\left(\tau ,t\right)\to {\bar{x}}^{\pm }\left(\tau \right)$ for the stationary profile of agents across ages. We also introduce ${\bar{x}}^{\pm }=\int_{0}^{\infty }d\tau \;{\bar{x}}^{\pm }\left(\tau \right)$. We wish to solve Equation (Equation 38) for this stationary profile ${\bar{x}}^{\pm }\left(\tau \right)$. For τ≠0, we have
(39)$$\begin{array}{c} \frac{\partial {\bar{x}}^{\pm }\left(\tau \right)}{\partial \tau }=-\left[a+p_{\tau }{\bar{x}}^{\mp }\right]{\bar{x}}^{\pm }\left(\tau \right), \end{array}$$
which yields
(40)$$\begin{array}{cc} {\bar{x}}^{\pm }\left(\tau \right) & ={\bar{x}}^{\pm }\left(0\right)\exp\left[-a\tau -{\bar{x}}^{\mp }\int_{0}^{\tau }ds\;p_{s}\right]. \end{array}$$

Now, using the boundary condition in Equation (Equation 38), one obtains the following expression for the stationary influx of voters into the states ±
(41)$$\begin{array}{cc} {\bar{x}}^{\pm }\left(0\right) & =\int_{0}^{\infty }d\tau {\bar{x}}^{\mp }\left(\tau \right)\left[a+p_{\tau }{\bar{x}}^{\pm }\right]. \end{array}$$ Substituting Equation (Equation 40) into Equation (Equation 41), and realising that the integral over τ becomes trivial, we obtain
(42)$$\begin{array}{c} {\bar{x}}^{+}\left(0\right)={\bar{x}}^{-}\left(0\right)\equiv \bar{x}\left(0\right). \end{array}$$ We note that ${\bar{x}}^{\pm }\left(\tau =0\right)$ describes agents that have newly arrived in state ±1 (and hence have age zero). Hence, this is the influx of agents into state ±1 in the stationary state. It is then clear why Equation (Equation 42) must hold: if the influxes into the two opinion states were different from each other, then there would be a net flow of agents from one state to the other, and the system would be non-stationary.

Next, integrating both sides of Equation (Equation 40) with respect to τ, and using ${\bar{x}}^{+}\left(0\right)={\bar{x}}^{-}\left(0\right)$, we arrive at the following closed equation for ${\bar{x}}^{+}$
(43)$$\begin{array}{cc} \frac{1}{{\bar{x}}^{+}}\int_{0}^{\infty }d\tau \;\exp\left[-a\tau -\left(1-{\bar{x}}^{+}\right)\int_{0}^{\tau }ds\;p_{s}\right] & =\frac{1}{1-{\bar{x}}^{+}}\int_{0}^{\infty }d\tau \;\exp\left[-a\tau -{\bar{x}}^{+}\int_{0}^{\tau }ds\;p_{s}\right]. \end{array}$$ For any given ageing profile $p_{\tau }$, Equation (Equation 43) can be solved for the deterministic fixed point ${\bar{x}}^{+}$. We note that the trivial solution ${\bar{x}}^{+}=1/2$ always exists. However, there are ageing profiles for which there are further non-trivial solutions.

We consider the specific example $p_{\tau }=\gamma /\left(t_{0}+\tau \right)$. Evaluating the integrals in Equation (Equation 43), we define
(44)$$\begin{array}{cc} I\left({\bar{x}}^{+}\right)\equiv \int_{0}^{\infty }d\tau \;\exp\left[-a\tau -{\bar{x}}^{+}\int_{0}^{\tau }ds\;p_{s}\right] & =\int_{0}^{\infty }d\tau \;{\left(1+\frac{\tau }{t_{0}}\right)}^{-\gamma {\bar{x}}^{+}}e^{-a\tau }\\ \; & =a^{-1}{\left(t_{0}a\right)}^{\gamma {\bar{x}}^{+}}e^{at_{0}}\Gamma \left(1-\gamma {\bar{x}}^{+},t_{0}a\right). \end{array}$$ One thus finds the fixed points ${\bar{x}}^{+}$ and ${\bar{x}}^{-}=1-{\bar{x}}^{+}$ by solving
(45)$$\begin{array}{c} \frac{1-{\bar{x}}^{+}}{{\bar{x}}^{+}}=\frac{I(x^{+})}{I(1-{\bar{x}}^{+})}. \end{array}$$ There is no closed-form analytical solution in this case, but Equation (Equation 45) can be solved numerically. We find (see Figure 5) that, for certain values of *a*, multiple solutions are possible. As *a* is increased, the non-trivial fixed points converge to the central value ${\bar{x}}^{+}=1/2$ and we are left with the usual coexistence stationary state. Importantly, the value of *a* at which the bifurcation occurs is independent of *N*, and the transition is now preserved in the thermodynamic limit N→∞.

#### 3.3.2. Fluctuations about the Steady State

Having found the deterministic fixed points of the noisy voter model with ageing, we now improve on this picture by including the stochastic fluctuations about these fixed points. Because the rates at which voters change from one opinion to the other are both concentration-dependent and non-Markovian, a Gaussian approximation to the noise along the lines of Ref. [33] is difficult to achieve. In the present case, we instead make an adiabatic approximation [39]. Specifically, we imagine that there is a separation of time scales between the time it takes for the stationary distribution of ages to be arrived at and the time between large changes in the population of opinions. More precisely, and as described below, we assume that the profile of ages in the population is stationary given the current distribution of agents $n^{\pm }\left(t\right)$ across the two opinion states ±1.

We begin by noting that the total rates at which voters of age τ switch from opinion ± to ∓ in the population at time *t* are given by
(46)$$\begin{array}{c} \Omega_{\tau }^{\pm }\left(t\right)=\left[a+p_{\tau }x^{\pm }\left(t\right)\right]n^{\mp }\left(\tau ,t\right). \end{array}$$ We stress that we use ± as a superscript for the rate for transitions from ∓ to ±.

We now regard the global quantities $x^{\pm }\left(t\right)=n^{\pm }\left(t\right)/N$ as slow variables to which the quantities $n^{\pm }\left(\tau ,t\right)$ are enslaved. One can then replace $n^{\pm }\left(\tau ,t\right)$ with their stationary averages, conditioned on the global variables $n^{\pm }\left(t\right)$. Specifically, $n^{\pm }\left(\tau ,t\right)$ satisfy (c.f. Equation (Equation 40))
(47)$$\begin{array}{cc} {\bar{n}}^{\pm }\left(\tau ,t\right) & =n^{\pm }\left(0,t\right)\exp\left[-a\tau -\frac{n^{\mp }\left(t\right)}{N}\int_{0}^{\tau }ds\;p_{s}\right]. \end{array}$$ From Equation (Equation 46), we obtain
(48)$$\begin{array}{c} \Omega^{\pm }\left(t\right)\equiv \int_{0}^{\infty }d\tau \;\Omega_{\tau }^{\pm }\left(t\right)=\int_{0}^{\infty }d\tau \left[a+p_{\tau }x^{\pm }\left(t\right)\right]n^{\mp }\left(\tau ,t\right)=n^{\mp }\left(0,t\right), \end{array}$$
where we have used Equation (Equation 47) in the last step, and the fact that the integral over τ can be carried out directly. Further, integrating both sides of Equation (Equation 47) with respect to τ, and using the definition of $I(x^{\pm })$ in Equation (Equation 44), we have $n^{\pm }\left(t\right)=n^{\pm }\left(0,t\right)I\left(x^{\mp }\right)$.
(49)$$\begin{array}{c} \Omega^{\pm }\left(t\right)=\frac{n^{\mp }\left(t\right)}{I[x^{\pm }\left(t\right)]}. \end{array}$$

The stationary state for $n^{\pm }$ can now be approximated from the one-step process for $n^{+}$. We have $n^{+}\to n^{+}-1$ with rate $\Omega^{-}\left(n^{+}\right)=n^{+}/I\left[\left(N-n^{+}\right)/N\right]$, and $n^{+}\to n^{+}+1$ with rate $\Omega^{+}\left(n^{+}\right)=\left(N-n^{+}\right)/I\left[n^{+}/N\right]$.

Using the well-known result for the WKB (or Eikonal) approximation of the stationary distribution of the master equation $P_{\mathrm{st}}\left(x^{+}\right)\propto \exp\left\{N\int^{x^{+}}dy\ln\left[\frac{\Omega^{+}\left(Ny\right)}{\Omega^{-}\left(Ny\right)}\right]\right\}$ [63], one finally obtains
(50)$$\begin{array}{c} P_{\mathrm{st}}\left(x^{+}\right)\propto \exp\left\{N\int^{x^{+}}dy\ln\left[\frac{\left(1-y)I(1-y\right)}{yI(y)}\right]\right\}. \end{array}$$ This approximation for the stationary state is verified in Figure 6. One notes that the maxima of the stationary distribution are given by the fixed points in Equation (Equation 43). Our approximation of the stationary distribution is valid in parameter regimes where there is only one fixed point. In the vicinity of the transition point and beyond, the adiabatic approximation no longer applies.

## 4. Discussion and Conclusions

In this work, we have studied an augmented version of the voter model, in which the rate at which individuals are persuaded to change their opinion depends on their age, i.e., on the time since the individual last changed opinion. Our analysis is based on numerical approaches and on analytical approximations. We outlined a modified version of Lewis and Shedler’s thinning algorithm, which allowed us to carry out efficient simulations of the system. We then went on to analytically characterise the deterministic dynamics (the analogue of rate equations in this non-Markovian system) for the voter model with ageing, but without the possibility for agents to change opinions spontaneously. Depending on the form of the ageing profile, we find both a power-law and exponential approach to consensus, as well as frozen states in which both opinion states remain in the population indefinitely.

Allowing for the possibility of spontaneous opinion changes alters the behaviour of the system. We demonstrated within a deterministic approximation that the fixed points then undergo a pitchfork bifurcation at a critical value of the propensity to spontaneously change opinions. Unlike in the conventional voter model (without ageing), the location of the transition point does not depend on the size of the population, and the transition remains even in the thermodynamic limit.

Although no simple master equation can be formulated for the model (due to its non-Markovian dynamics), we were also able to go beyond a deterministic description, providing an approximation to the stationary distribution of fluctuations about the fixed points. Depending on the rate of spontaneous opinion changes, we find unimodal or bimodal distributions for the number of agents in either opinion state.

The use of continuous ageing has several advantages in comparison to earlier models with discrete ageing. First, it is a much more natural way to describe the ageing process—changes in the proclivity of individuals to alter their opinions do not occur suddenly, but rather vary gradually. Further, the expressions that are obtained from the analytical treatment are more compact. For example, we obtain a closed equation for the approach to consensus in Equation (Equation 17), whereas two coupled equations were required in Ref. [38]. With this equation, we were able to find, for several ageing profiles, whether or not the system reaches asymptotically the consensus state and at what rate.

The types of equations that we encounter in this work are familiar from other problems in which memory effects are important. For example, the convolution with the memory kernel in Equation (Equation 17), which resulted from our elimination of the age variables, is also found in the context of anomalous diffusion [33,56], and in systems with distributed delay [26,28]. Our work therefore connects the voter model with ageing with this existing literature. We also showed that the memory kernel can be traded for a fractional derivative for survival time distributions with a power-law tail with an exponent γ between zero and one. This is a general trait in non-Markovian systems [34,35].

One feature that sets the voter model with ageing apart from many existing systems with non-Markovian dynamics is the fact that the rate with which agents change opinion depends on the fraction of voters in the opposite state. Hopping rates in systems with anomalous diffusion, in contrast, typically do not depend on the concentration of other substances in the system [60]. Similarly, delayed recovery in a model of epidemics does not depend on, say, the number of susceptible or recovered agents in the population. The concentration dependence in the reaction rates adds a layer of complexity in the voter model with ageing. It is the combination of non-Markovianity and concentration dependence that meant that we had to linearise about the absorbing state in order to eliminate the age variable τ. No such linearisation was required to eliminate the age variables in, for example, Refs. [26,28,33,56,57].

We envisage that the approaches developed here could be useful for a variety of other problems. The thinning algorithm by Lewis and Shedler [49] is somewhat undervalued in our opinion and can be modified for systems beyond those involving single-species Poisson processes, for which it was originally conceived. We have here shown how the algorithm can be used for systems in which the reaction rates at a given point vary in time due to a dependence on the state of the system at an earlier time. We also anticipate that the analytical methods that we used could be applied in other systems. Fluctuations in subdiffusive systems or gene regulatory circuits are often treated using a Gaussian approximation [28,29,33]. One future avenue might be to try to eliminate the equivalent of the ageing variable in these systems along the lines of what we have done in Section 3.3.2. One could then try to characterise the stationary distribution of these models based on a reduced Markovian birth–death process. As a further line of future work, our method for studying the approach to consensus could also be re-purposed for the approach to absorbing states in more general non-Markovian models. The approach to fade-out in models of an epidemic could be an example.

As in many models in sociophysics, it is relevant to study the effect of the network of interactions. This has been studied numerically for the voter model with ageing both for the characteristic decay to consensus [44] and the nature of the phase transition in the case with noise [40] for different random network structures (regular 2,3,4-dimensional, Barabasi–Albert, Erdos–Renyi, etc.). It would be interesting to combine the analytical techniques used in this study with recently developed approximate methods specifically designed to study the dynamics of agent-based models on complex networks (see [64] for a recent approach and a revision of existing methods). 

## Figures and Tables

**Figure 1 entropy-24-01331-f001:**
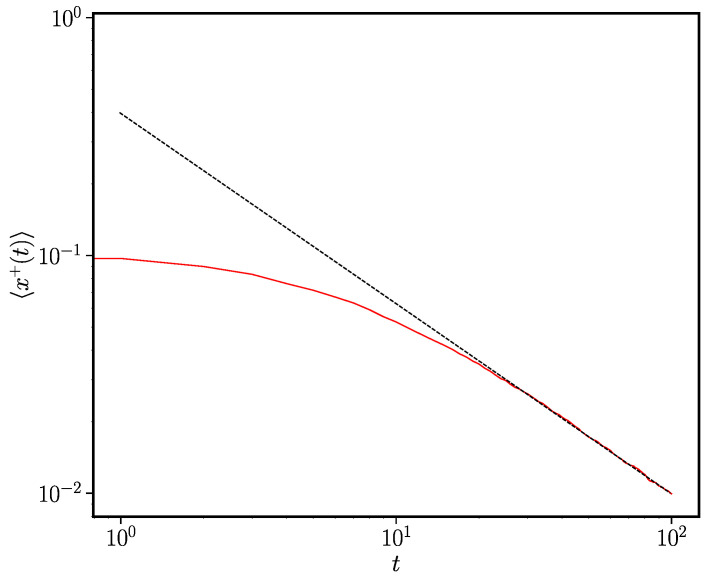
The ageing-induced power-law approach to consensus. Individual-based simulations of the ageing voter model were performed using the method given in Section 2.2 for the case where $p_{\tau }=\gamma /\left(t+t_{0}\right)$. The solid red line is the mean $\langle x^{+}\left(t\right)\rangle$ averaged over 1000 trials and the dotted black line is $x^{+}\left(t\right)=t^{-\gamma }$. The remaining system parameters were γ=0.8, N=100, $t_{0}=0.8$.

**Figure 2 entropy-24-01331-f002:**
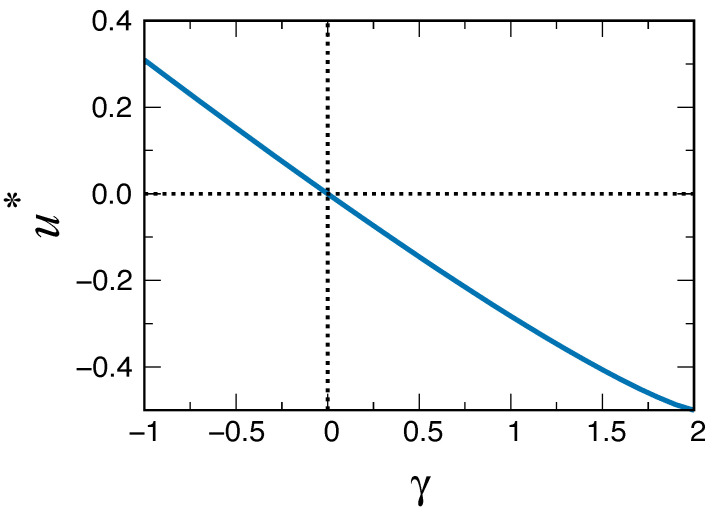
The location of the pole $u^{*}$ of the right-hand side of Equation (Equation 30), given by the solution of $\hat{\Psi }\left(u^{*}\right)=\frac{1}{p_{\infty }}$, as a function of γ for $t_{0}=2.0$, $p_{\infty }=0.5$. Note that the sign of $u^{*}$ matches that of γ (see the text for the discussion of this result).

**Figure 3 entropy-24-01331-f003:**
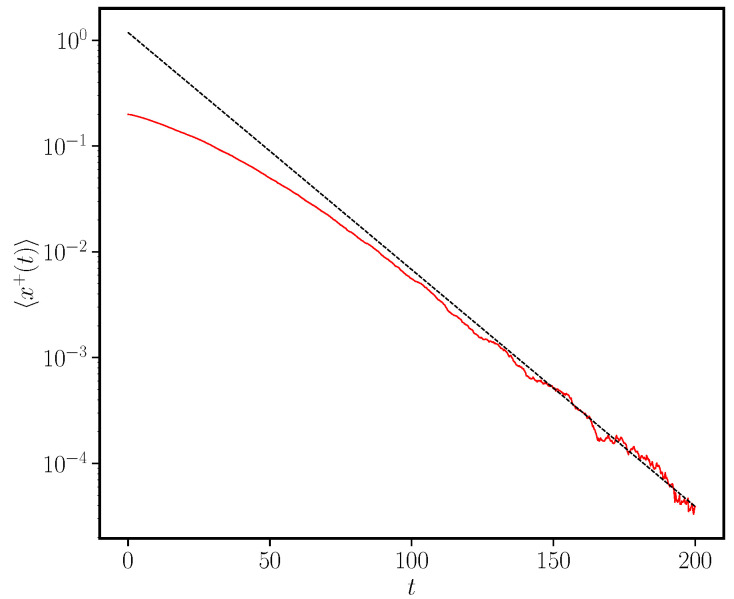
We plot (solid line) the evolution of the average density $\langle x^{*}\left(t\right)\rangle$ of agents holding the +1 opinion as a function of time *t* using the functional form for the activation rate given by Equation (Equation 28) with $p_{\infty }=0.5,\;t_{0}=0.8,\;\gamma =0.1$. The results have been averaged over 100 realisations of the stochastic dynamics starting at $x^{*}\left(0\right)=0.2$. The dashed line is the theoretical prediction of an asymptotic exponential decay $x^{+}\left(t\right)∼e^{u^{*}t}$ with a value of $u^{*}=-0.0517$ given by the solution of $\hat{\Psi }\left(u^{*}\right)=\frac{1}{p_{\infty }}$, the pole of the right-hand side of Equation (Equation 30).

**Figure 4 entropy-24-01331-f004:**
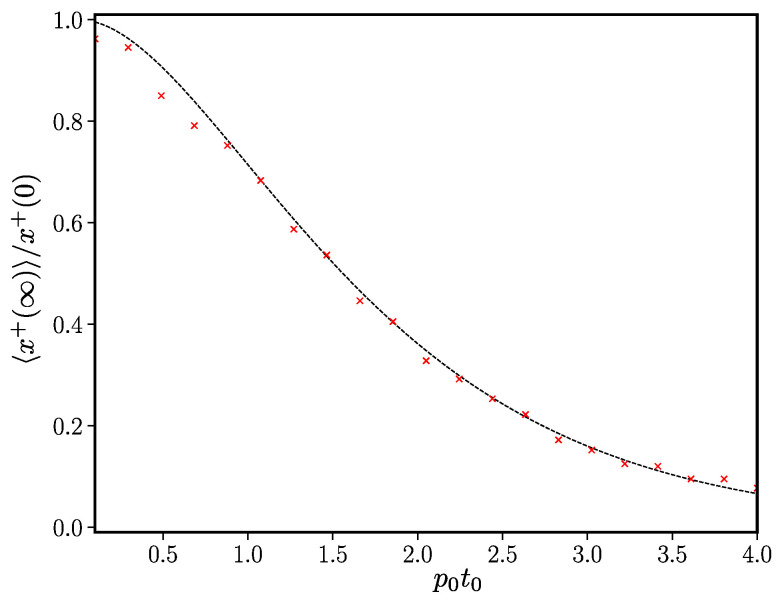
The final frozen state in the case of exponentially decaying transition rate. Results of individual-based simulations of the ageing voter model were performed using the Lewis and Shedler thinning method (see Section 2.2) for the case where $p_{\tau }=p_{0}e^{-\tau /t_{0}}$. The red points are the simulation results for the mean $\langle x^{+}\left(t\right)\rangle$ averaged over 100 trials and the dotted black line is the series in Equation (Equation 36) truncated after 100 terms. The remaining system parameters were $t_{0}=1$, N=1000.

**Figure 5 entropy-24-01331-f005:**
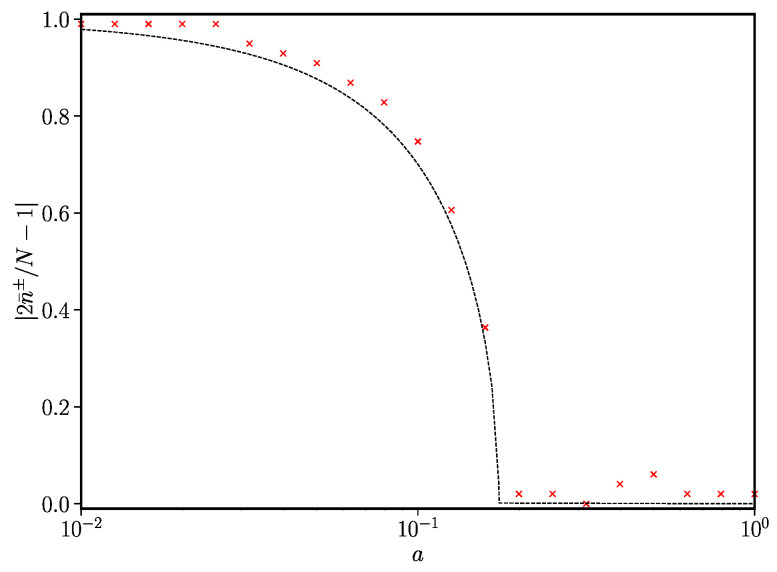
Ageing-modified noise-induced phase transition. Here, $t_{0}=1$, γ=2, N=100. Coloured crosses represent the modal value of the magnetisation $m=|x^{+}-x^{-}|=|2x^{\pm }-1|$ obtained from simulations using the method detailed in Section 2.2. The dashed line is the solution of Equation (Equation 45).

**Figure 6 entropy-24-01331-f006:**
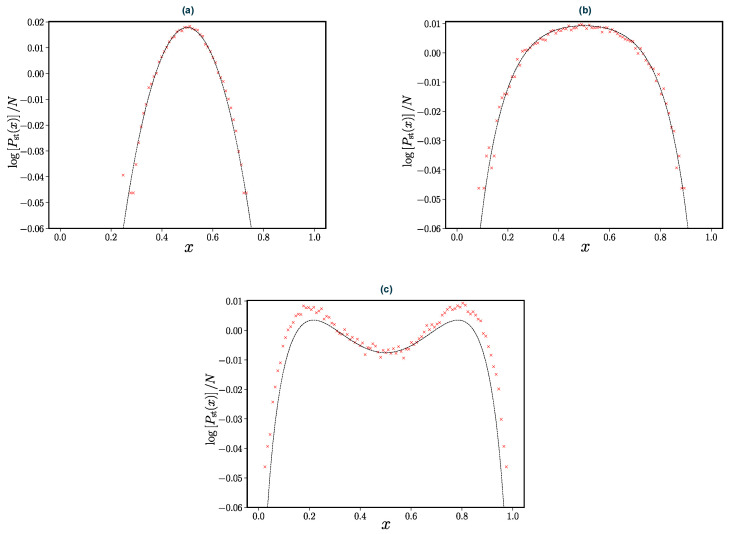
The stationary distribution of the concentration $x^{+}$ for various values of the noise strength ((**a**) a=1, (**b**) a=0.2, (**c**) a=0.126). The age-dependent rate in Equation (Equation 18) was used with fixed parameters $t_{0}=1$, γ=2 and N=100. Coloured crosses are the results of simulations using the Lewis and Shedler thinning method discussed in Section 2.2 and the dashed line is given by Equation (Equation 50).

## Data Availability

All relevant data are included within the paper.

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
