# Peer review of "Analytical and Numerical Treatment of Continuous Ageing in the Voter Model"

_entropy, 2022, doi:10.3390/e24101331_

Round 1

Reviewer 1 Report

The authors study a non-Markovian version of the voter model, in which the switching probability of the agents depend on the time since they last switched. There are interesting analytical and numerical treatments, in the mean field limit, which could be useful in realistic situations.

I believe that the manuscript warrant publication. However, I would expect the authors to consider the following points:

1. The term "aging" is a bit confusing here. Once the opinion is switched, "age" value is reset. So, its not quite "aging" after all (no absolute scale of time), it is rather like "friction" (contact points becoming stronger with time).

The authors should clarify that it is not quite "aging".

2. It is very interesting to see the freezing effect of Fig. 4. It will be interesting to elaborate how does the saturation value of frozen fraction depend upon the rate "tau". It is also very interesting, possibly for future direction, how does the frozen fraction vary in lower dimensional cases. To me this looks like a "quenching schedule", so it would be interesting to possibly relate the defect density (say domain wall density) with the "quenching rate" tau. 

3. The fully connected graph is a very idealized situation. It would be interesting if the authors could comment on the effect of topology (e.g., scale free network, lowed dimensional lattices) on the results.

Reviewer 2 Report

The authors study a modified voter model in which the switching rate
of the agent depends of the difference in time between two successive
switchings, which is called age
To this end they set up a non-markovian dynamics
with rates that depends on the concentration. The resulting model
is studied by means of simulation as well as by analytical methods.
They demonstrated how the absorbing state is reached for long
times. Three cases are discussed according to the dependence of the
switching rate on the age. They also consider the effects of
spontaneous change of opinion, which is the noisy voter model
with continuous ageing. In this last case, a continuous transition
between coexistence and consensus phases is found. Although
the approach they use cannot be described by a master equation,
the authors show how the stationary probability distribution function
can be obtained.
The paper is an interesting work on the variants of the voter model.
It is well written and the conclusions follows from the
premises of the model.

Author Response

We thank the reviewer for his/her comments. There are no specific points to address in this review.